# Low Alanine Aminotransferase Cut-Off for Predicting Liver Outcomes; A Nationwide Population-Based Longitudinal Cohort Study

**DOI:** 10.3390/jcm8091445

**Published:** 2019-09-11

**Authors:** Jin Hwa Park, Jun Choi, Dae Won Jun, Sung Won Han, Yee Hui Yeo, Mindie H. Nguyen

**Affiliations:** 1Department Internal Medicine, Hanyang University, Seoul 133-792, Korea; pjh6718@hanmail.net; 2Department of Fusion Data Analytics, School of Industrial Management Engineering, Korea University, Seoul 136-701, Korea; jun.choi@outlook.kr; 3Division of Gastroenterology and Hepatology, Stanford University Medical Center, Palo Alto, CA 94305, USA; yeehuiy@stanford.edu (Y.H.Y.); mindiehn@stanford.edu (M.H.N.)

**Keywords:** alanine aminotransferase, mortality, upper limit of normal, liver function tests

## Abstract

Background and aim: Recent practice guidelines suggest healthy normal alanine aminotransferase (ALT) levels should be less than 30 U/L for males and 19 U/L for females. We tried to validate the prediction power of the “low cut off” for liver related outcomes in the general population. Methods: A total of 426,013 subjects were followed up for 10 years using the National Health Screening Cohort database. Prediction ability of long term mortality and liver related outcomes between conventional (<40 U/L in men and women) and low (<30 U/L in men and <19 U/L in women) ALT cut-off values were compared. Results: Both conventional and low ALT cut-offs predicted liver related unfavorable outcomes in Kaplan-Meier analysis. Following adjustment for age, body mass index, smoking, exercise, alcohol consumption, fasting blood glucose, and cholesterol via multivariate Cox regression, abnormal ALT using new ‘low ALT cut off’ was a significant independent predictor for liver-related mortality, HCC, and decompensated liver events. When the low cut-off criteria were added to the prediction model, the ability to predetect liver-related hard outcomes significantly increased in both men and women (*p*-values < 0.0001). The C-index values for predicting liver-related adverse events were the same in both ALT cut-offs, after adjusting confounding factors (C index value: 0.73~0.88). Conclusions: New low ALT cut-off showed good prediction power for liver related unfavorable outcomes.

## 1. Introduction

The alanine aminotransferase (ALT) test has been widely used to screen, detect, and follow up on the status of liver diseases. ALT reference values for the upper limit of normal (ULN) vary greatly depending on laboratories, with ULN ranging from 35–79 U/L in men and 31–55 U/L in women. Although there is no unified ALT ULN standard established, ALT < 40 U/L in men and women is generally used as the conventional cut-off value of ALT. However, there have been many suggestions that the current conventional ALT cut-off is too high. More recently, several professional management guidelines have suggested a lower cut-off for a “true” ULN of ALT [1,2].

In a landmark study, Prati et al. was the first to report that ALT ULNs in healthy populations without viral hepatitis or metabolic disease (no evidence of alcoholic liver disease, or diabetes, and normal BMI and waist circumference) should be 30 U/L in men and 19 U/L in women. Subsequently, several cross-sectional studies also reported lower ULNs for ALT in healthy individuals [2,3,4]. For instance, ULNs of ALT were proposed as 29 U/L in men and 22 U/L in women using the National Health and Nutrition Examination Survey of the United States [2]. In another cross-sectional study, ULNs of ALT in 1105 biopsy-proven healthy liver donors were 33 U/L in men and 25 U/L in women [3]. The American College of Gastroenterology proposed ULNs of 29–33 U/L for men, and 19–25 U/L for women as the “true healthy normal ALT” [5]. 

However, most studies to date were cross-sectional studies, while data from longitudinal studies to demonstrate a correlation between different ALT cut-off levels (conventional vs. lower) with development of liver-related morbidity and mortality are limited and/or conflicting. Arndt et al. showed that “high” aspartate aminotransferase (AST > 18 U/L) was associated with a three-times higher risk of all-cause mortality but not a lower level of ALT elevation [6]. In another study conducted in Korea, lower cut-off of aminotransferases correlated with mortality in men but not in women [7]. Ruhl et al. were the first to demonstrate that the lower ALT cut-off (<0 U/L in men or <19 U/L in women) can predict liver disease mortality in the U.S. general population [8]. However, the study sample size was relatively small (14,950 subjects), and only 34 liver-related mortalities developed during the study observation period. Moreover, this study did not evaluate liver-related morbidity (hepatocellular carcinoma [HCC] and decompensated liver disease). As a result, additional data on liver-related mortality and morbidity outcomes with long-term follow-up of subjects with lower ALT levels are needed especially in the Asian population. 

Therefore, the aim of our study was twofold. The first was to compare liver related outcomes of mortality, HCC and decompensated events between the low levels of ALT of 30 U/L for men and 19 U/L for women compared to the conventional level of <40 U/L for men or women. The second was to identify the optimal cut-off value to predict liver related mortality, HCC, and decompensated events in the general South Korean population using a longitudinal study design.

## 2. Methods

### 2.1. Study Population and Data Source

In South Korea, there are several databases available to obtain patient level data. Our study was based on National Health Insurance Service-National Health Screening Cohort (NHIS-HEALS) registry which consists of three databases (personal information database, prescription and diagnosis database, and health screening database). The current study used the NHIS-HEALS registry to select a random 10% sample of 5.15 million qualified NHIS patients aged 40 to 79 out of the total general health screening examinees in the system from 2002 to 2013. They had health checkups in 2003–2004 and were followed up to 2013 [9]. In order to assess for pre-existing diseases, we excluded data from 2002-further inclusion and exclusion criteria are described below subjects.

The access to the registry is provided to researchers after approval by the Institutional Review Board (IRB) of the NHIS and the local IRB (https://nhiss.nhis.or.kr/bd/ab/bdaba000eng.do). The study was approved by the Hanyang University IRB (HYI-17-073-1) and the NIHS IRB. Informed consent was waived because of the retrospective nature of the study. All process was performed in accordance with the relevant guidelines and regulations by both IRB and HIRA. We used an open data source, so the informed consent was waived.

### 2.2. Inclusion Criteria

‘General population’ was defined as those without known malignancy of any type, known liver disease, or diseases known to affect liver enzymes who had a checkup in the years 2003–2004.

### 2.3. Exclusion Criteria

Individuals with pre-existing liver disease or serious medical disease, which can affect liver enzymes, and individuals with high-risk liver diseases were excluded. The high-risk group included those diagnosed with cancer (ICD-10 code: C00–C99), hepatitis B (ICD-10 code: B16.2, B19.11, B18.1, B16.9, B19.10, B18.1), hepatitis C (ICD-10 code: B17.11, B18.2, B17.10, B18.2, B19.20, B19.21), alcoholic liver disease (ICD-10 code: K70.0, K70.10, K70.31, K70.2, K70.30, K709), non-alcoholic fatty liver disease (K76.0, K76.89, K76.9, K74.1), and other chronic diseases (code: K75.4, K83.0, K74.3, K74.5, K74.4, E88.01, E83.110, E83.111, E83.118, E83.119, E83.00, E83.01, E83.09). In addition, for more accurate analysis we removed missing values (22,242 subjects) and ALT outliers (10,577 subjects >99 percentile, ALT <9 U/L or >95 U/L). 

### 2.4. Demographic and Clinical Parameters

The following data were collected: age, height, body weight, blood pressure, fasting glucose, cholesterol, ALT, aspartate aminotransferase (AST), and gamma-glutamyl transferase (GGT). Blood tests taken at health check-up clinics were performed 8 h after fasting. Data from patient questionnaires on smoking habits and alcohol consumption were also collected.

### 2.5. Primary and Secondary End-Points

The primary end-point was the ability of low ALT cut-off criteria to determine liver-related mortality and morbidity (HCC and decompensated liver events). The secondary end-point was the ability of conventional cut-off versus the low ALT cut-off to predict the incidence of HCC and decompensated liver events in a longitudinal manner.

### 2.6. Operational Definitions

#### 2.6.1. Mortality

The mortality information was obtained from external data provided by the Statistics Korea (a central organization for statistics under the Ministry of Strategy and Finance of the Republic of Korea, kostat.go.kr). Causes of death were analyzed using the Korean Standard Classification of Diseases (KCD) codes of the Korean Standard Classification of Diseases and Causes of Death provided by the Statistics Korea. The Statistics Korea has developed the KCD as recommended by the World Health Organization (WHO), and records the Korean Standard Classification of Diseases and Causes of Death to ensure accuracy in the comparison of diseases and death-related statistical data since 1952.

#### 2.6.2. Hepatocellular Carcinoma

HCC (diagnosis code: C22.0) was diagnosed by biopsy or by the presence of any hypervascular lesion with arterial enhancement and portal or delay washout on dynamic contrast enhanced CT and MRI [10]. There are strict standards for HCC diagnosis in Korea, since patients with a C22.0 diagnosis code receive reimbursement for 95% of all their medical expenses from the NHIS. 

#### 2.6.3. Decompensated Liver Disease

Decompensated liver events included uncontrolled ascites, variceal bleeding, hepatic encephalopathy, and hepatorenal syndrome. The operational definition of uncontrolled ascites included subjects receiving paracentesis of more than 3 L (C8050, C8051, and Q2470) and a serum albumin level less than 3.0 g/dL. The definition of variceal bleeding included subjects who received sclerotherapy or endoscopic ligation treatment (Q2430, Q2431, Q2432, Q2433, Q2434, Q2435, Q2436, Q2437, Q2438, Q7631, Q7632, Q7633, and Q7634) or those who were treated with vasopressin, terlipressin, somatostatin, or octreotide. Subjects who were prescribed a lactulose enema (M0076) during admission were considered to have hepatic encephalopathy, and subjects who were prescribed terlipressin and albumin at the same time during admission were considered to have hepatorenal syndrome.

## 3. Statistical Analysis

The student t-test and chi-square tests were used to investigate the differences in the clinical and demographic data between groups. Kaplan-Meier methods were used to estimate cumulative incidence of HCC, decompensated liver events, and liver-related death, with comparisons between subgroups by ALT cut-offs carried out using the log-rank test.

Multivariate Cox regression models were used to estimate hazard ratios relating ALT levels with various outcomes (HCC, decompensated liver events, and liver-related mortality), with adjustment for age, body mass index (BMI), smoking, exercise, alcohol consumption, fasting blood glucose, and cholesterol levels. AST and gamma-glutamyl transpeptidase (GGT) were not included in the Cox regression model, since they demonstrated a high correlation with ALT. Concordance statistic (C-index) value was calculated to determine both ALT cut-offs’ ability to predict liver related unfavorable outcomes. Prediction power expressed from 0.5 to 1.0. A value of 1.0 means perfect predictability. The C-index were derived based on regression analysis.

In addition, a likelihood ratio (LR) test was carried out to evaluate if the predictability of liver-related mortality, HCC, and decompensated liver events could be enhanced when low cut-off values were compared with a control model. Definition of ‘control model’ is a basic and simple model for predictability of liver-related mortality, HCC, and decompensated liver events using universal variables. The universal variables in the control model were age, BMI, smoking, exercise, alcohol consumption, fasting blood glucose, and cholesterol for predicting liver-related mortality, HCC, and decompensated liver events. The low cut-off criteria were then added to the prediction model to assess the prediction model’s ability to detect liver-related mortality, HCC, and decompensated liver events with the low ALT cut-off criteria.

Finally, the best cut-off values and their predictive accuracy for each of the liver outcomes of interest were investigated via the area under the receiver-operating curves (AUROC) and associated 95% confidence intervals (CIs). Sensitivity and specificity were also calculated. Best cut-off values calculated by R package (https://cran.r-project.org/web/packages/OptimalCutpoints). All analyses were conducted using SAS software (version 9.4; SAS Institute, Inc., Cary, NC, USA).

## 4. Results

### 4.1. Baseline Characteristics 

A total of 338,216 subjects who did not have any liver-related KCD codes within the previous one year were enrolled, out of the 426,013 health screening examinees in 2003–2004 (Figure 1). The mean age of the cohort was 52.19 (9.46 SD) years old, BMI was 23.96 (2.95 SD) kg/m^2^, and the mean AST and ALT were 25.47 (11.29 SD) U/L and 24.13 (12.68 SD) U/L, respectively (Table 1). Compared to female subjects, male subjects had significantly higher mean ALT levels (27.04 vs. 23.56 U/L), GGT (47.00 vs. 20.82 U/L), proportion of smokers (41.17% vs. 2.50%) and non-alcohol consumers (35.41% vs. 82.63%) (*p* < 0.0001 for all comparisons). 

### 4.2. Long-Term Adverse Liver Outcomes

During follow-up (mean follow-up 9.83 years, standard deviation 1.29 years), there were 360 liver-related deaths, 28,363 cases of HCC, and 1048 decompensated liver events. 

### 4.3. Unfavorable Liver Related Outcomes According to ALT Levels 

In both genders, higher ALT levels were associated with a significant increase in liver-related mortality, HCC, and decompensated liver events even after adjustment for age, gender, smoking, alcohol consumption, exercise, and blood glucose (Figure 2). Notably, the increase appeared to occur for all outcomes at approximately 30 U/L for men and 20 U/L for women.

### 4.4. Both Conventional and Low ALT Cut-Offs Predicted Liver Related Unfavorable Outcomes in Kaplan-Meier Analysis

By both low ALT cut-off (≤30 U/L in men and ≤19 U/L in women) and conventional ALT cut-off (ALT ≤ 40 U/L for both men and women), subjects with higher ALT levels had significantly higher liver-related mortality, incidence of HCC, and decompensated liver events (*p* < 0.0001 for both cut-offs in all three endpoints) (Figure 3, Appendix A). Both ALT cut-offs clearly discriminate three long term liver-related unfavorable outcomes.

### 4.5. Both Conventional and Low ALT Cut-Offs Predicted Liver Related Unfavorable Outcomes in Cox Regression Analysis

Following adjustment for age, body mass index, smoking, exercise, alcohol consumption, fasting blood glucose, and cholesterol via multivariate Cox regression, ALT > 30 U/L for men and >19 U/L for women were significant independent predictors for liver-related mortality, HCC, and decompensated liver events (Table 2). The adjusted hazard ratios for liver-related mortality by the conventional ALT cut-off (40 U/L) were 6.96 (95% CI: 5.48–8.86) in men and 12.06 (95% CI: 7.43–19.58) in women. For the low ALT cut-offs (30 U/L for men and 19 U/L for women), the adjusted hazard ratio for liver-related mortality was 5.18 (95% CI: 4.04–6.63) in men and 4.78 (95% CI: 2.76–8.27) in women. Similarly, multivariate hazard ratios for the prediction of HCC and decompensated liver events were also significantly elevated in both men and women with the lower ALT cut-offs of 30 U/L and 19 U/L, respectively.

### 4.6. Both Conventional and Low ALT Cut-Offs Enhanced the Likelihood Ratio of Liver Related Unfavorable Outcomes

The likelihood ratio (LR) test was carried out to evaluate if the predictability of liver-related mortality, HCC, and decompensated liver events were enhanced when low cut-off values were compared with the control model. When the low cut-off criteria were added to the prediction model, the prediction model’s ability to detect liver-related mortality, HCC, and decompensated liver events significantly increased in both men and women (*p*-values for all 6 models were <0.0001) (Figure 4A). 

### 4.7. Concordance Statistic (C-index) Value between Conventional and Low ALT Cut-Off Was Not Different

Concordance statistic (C-index) means degree of fit for a logistic regression model in binary outcomes. The C-index values for predicting liver-related adverse events were the same in both ALT cut-offs, after adjusting for age, BMI, smoking, exercise, alcohol consumption, fasting blood glucose, and cholesterol (C index value: 0.73~0.88) (Table 3). There was no difference of 95% confidence intervals of C-index value between conventional and low ALT cut-offs.

### 4.8. Best Cut-Off Values of ALT According to Long Term Liver Related Hard Outcomes

Receiver-operating characteristic curves (ROC) were constructed to identify the optimal ALT. Best cut-offs calculated by R package (Figure 4B). In men, area under the curve (AUC) of ALT for liver-related mortality was 0.71, and best cut-off point was 31.5 U/L. AUC of decompensated liver events was 0.65, and best ALT cut-off point was 33.5 U/L. AUC of HCC was 0.51, and best ALT cut-off point was 16.5 U/L. In women, area under the curve (AUC) of ALT for liver-related mortality was 0.76, and best cut off point was 24.5 U/L. AUC of decompensated liver events was 0.64, and best ALT cut-off point was 24.5 U/L. AUC of HCC was 0.54, and best ALT cut-off point was 24.5 U/L.

## 5. Discussion

In our study, when comparing the conventional level for the upper level of normal for ALT (<40 U/L) to the lower recommended ALT levels (30 U/L for men and 19 U/L for women), our data showed that the low ALT cut-off values predicted the risk of liver-related mortality, HCC, and decompensated liver events in the general South Korean population as well as the conventional level. There was no difference in the C-index values between the conventional and low ALT cut-offs after adjusting for confounding variables. Furthermore, the C-index results for the low cut-off values in predicting liver related unfavorable outcomes ranged between 0.73–0.88. However, we did find that the optimal cutoff values for this South Korean general population varied for men and women and the disease state under investigation. For men, an ALT value of 31.5 had the best predictive ability for liver related mortality while a level of 33.5 was best for decompensated events and 16.5 for HCC. For women, an ALT value of 24.5 was consistent for all three outcomes. On the other hand, the ALT level did not provide a good index for predicting the occurrence of HCC as shown when the AUROC’s for both men and women were approximately 50%. This finding should not be surprising as it has been established that cirrhosis is the strongest predictor for HCC and when ALT is investigated by itself it does not reflect the chance of developing HCC [11]. 

On the other hand, our optimal results closely follow the recently released 2018 HBV guidance by the American Association for the Study of Liver Diseases, which recommended a ULN for ALT of 35 U/L for men and 25 U/L for women [12] compared to ULNs of 30 U/L for men and <19 U/L for women in their 2016 version [2]. Our study demonstrates that these cut-off values are excellent levels to use when determining liver-related morbidity, liver cancer, and decompensated liver events, extending the utility of these values to the South Korean population. 

Furthermore, the strength of our findings showed that higher levels of ALT were better at discriminating those who would develop the liver adverse outcomes. This is important as a recent concern was expressed that very low ALT levels may predict higher overall mortality [13,14]. One meta-analysis showed that extremely low ALT levels were associated with higher overall mortality, such that there was an inverse association between serum ALT levels and all-cause mortality in the elderly [15]. In fact, another study found that ALTs less than 20 U/L for individuals older than 40 years of age were associated with increased overall mortality [14,16]. Therefore, careful interpretation of ALT levels within an appropriate clinical context is warranted in routine practice to avoid over or under testing. 

Our results closely follow a recent study also conducted in South Korea which investigated ALT levels in relation to liver related mortality. These investigators found that the optimal ALT levels were >34 U/L for men and >30 U/L for women [13]. However, our results differ in that Shim et al. were investigating the optimal levels for patients with hepatitis B virus while our study was for the general population such that our results may be more helpful to the general practitioner when evaluating patients for liver disease.

A major strength of our study is its population-based longitudinal cohort study design and inclusion of the largest number of subjects to date in the investigation of optimal ALT cut-off criteria. In addition, to the best of our knowledge, this is the first and largest study demonstrating the relationship between the lower ALT cut-offs of 30 U/L for men and 19 U/L for women and the development of liver-related morbidity and mortality in an average-risk population. Combined, these strengths allowed us to provide the general practitioner with guidance when working with a patient at average risk for liver disease as to whether further testing is warranted at what ALT level. 

Our study has limitations. First, despite our large sample size and long follow-up time, the number of liver related mortalities was low. However, this was expected, as our cohort was carefully designed to include only average risk or a general population. The number of events was still sufficient for statistical significance for the major analyses of interest. Second, ALT testing was performed at clinical laboratories throughout Korea instead of a central laboratory, which may have introduced some measurement bias. However, our sample size was very large and thus, any bias introduced may have been mitigated. Finally, data on alcohol consumption was obtained through a self-report questionnaire, which tends to underestimate actual alcohol consumption. 

In conclusion, we found that though the low ALT cutoff values (30 U/L for men, 19 U/L for women) do predict the risks of liver-related mortality, HCC, and decompensated liver events, the optimal ALT values for the average risk person in South Korea closely track with the new AASLD recommendations of an ALT of 35 for men and 25 for women. These findings will help the general practitioner in determining their care plan when working with the South Korean general population. 

## Figures and Tables

**Figure 1 jcm-08-01445-f001:**
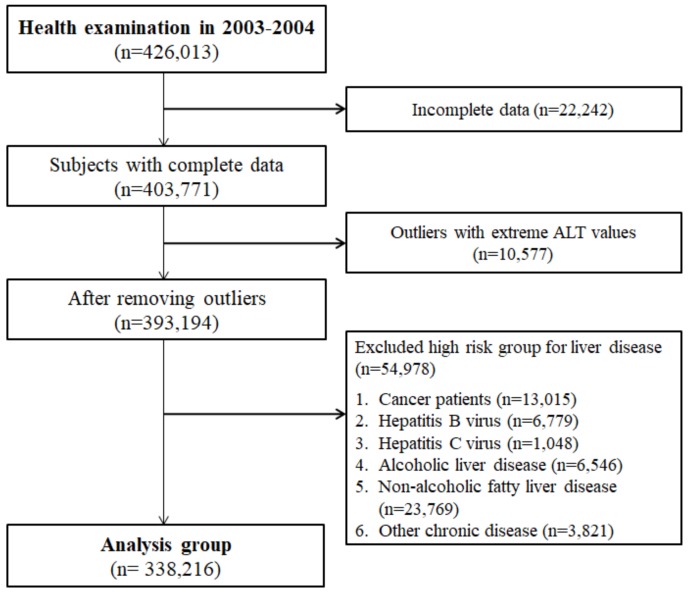
Patient enrollment flow chart.

**Figure 2 jcm-08-01445-f002:**
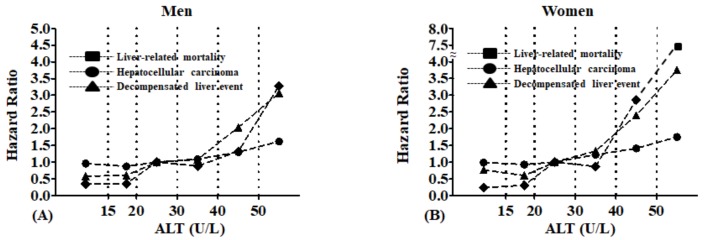
Multivariate hazard ratios relating alanine aminotransferase levels to liver-related mortality, hepatocellular carcinoma, and decompensated liver events according to Cox regression (adjusted for age, body mass index, smoking, exercise, alcohol consumption, fasting blood glucose, and cholesterol levels): (**A**) Men and (**B**) Women.

**Figure 3 jcm-08-01445-f003:**
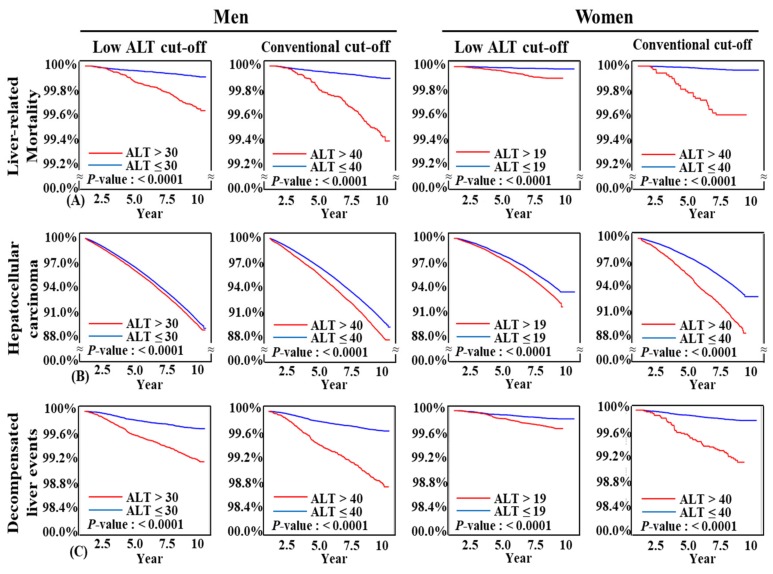
Kaplan-Meier curves for development of liver outcomes according to alanine aminotransferase levels: (**A**) Liver-related mortality; (**B**) hepatocellular carcinoma; and (**C**) decompensated liver events.

**Figure 4 jcm-08-01445-f004:**
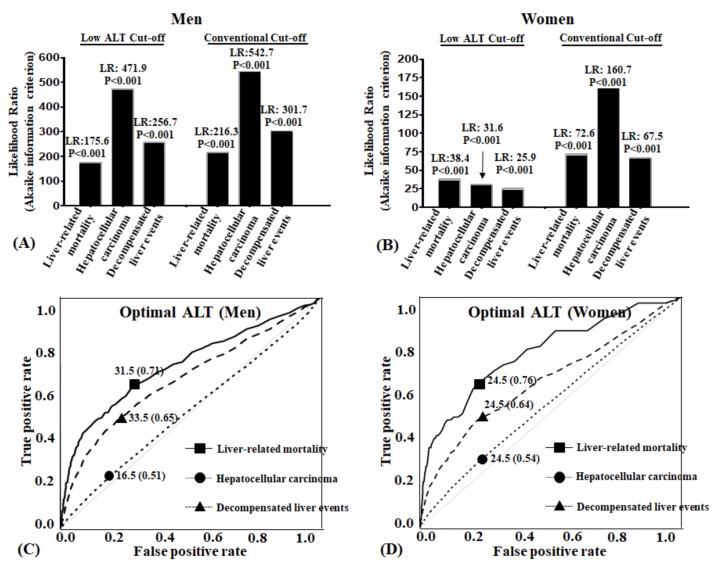
Likelihood ratio of mortality and morbidity and receiver operating characteristic curve according to ALT reference variables: (**A**,**B**) Cox proportional hazard regression models for likelihood ratio (LR) for liver outcomes according to two different alanine aminotransferase (ALT) cut-off values. All adjusted for age, BMI, smoking, exercise, alcohol consumption, fasting blood glucose, and cholesterol. LR, likelihood-ratio statistic: ΔG2 = −2 log L from reduced model(−2 log L from current model). Reduced model is the control model. Current model is the cut-off model; (**C**,**D**) Optimal cut-off values of alanine aminotransferase for adverse liver outcomes.

**Table 1 jcm-08-01445-t001:** Baseline characteristics.

Variable	All (*n* = 338,216)	Men (*n* = 185,510)	Women (*n* = 152,706)	*p*-Value ^a^
**Age (years)**	52.19 ± 9.46	51.36 ± 9.19	53.21 ± 9.68	<0.0001
**Body mass index, kg/m** ^**2**^	23.96 ± 2.95	23.96 ± 2.83	23.96 ± 3.08	0.5004
**AST (U/I)**	25.47 ± 11.29	27.04 ± 12.47	23.56 ± 9.31	<0.0001
**ALT (U/I)**	24.13 ± 12.68	27.18 ± 13.66	20.42 ± 10.21	<0.0001
**GGT (U/L)**	35.18 ± 44.68	47.00 ± 54.83	20.82 ± 19.82	<0.0001
**Cholesterol (mg/dL)**	200.31 ± 37.41	198.20 ± 36.49	202.90 ± 38.35	<0.0001
**Glucose (mg/dL)**	97.73 ± 31.60	99.45 ± 32.94	95.64 ± 29.77	<0.0001
**Smoking status**				
**Non-smoker**	227,928 (67.39)	80,405 (43.34)	147,523 (96.61)	<0.0001
**Ex-smoker**	30,105 (8.90)	28,737 (15.49)	1368 (0.90)	
**Current smoker**	80,183 (23.71)	76,368 (41.17)	3815 (2.50)	
**Alcohol consumption**				
**None**	191,865 (56.73)	65,682 (35.41)	126,183 (82.63)	<0.0001
**Seldom**	52,232 (15.44)	36,282 (19.56)	15,950 (10.44)	
**Sometimes**	56,986 (16.85)	49,024 (26.43)	7962 (5.21)	
**Often**	23,138 (6.84)	21,656 (11.67)	1482 (0.97)	
**Always**	13,995 (4.14)	12,866 (6.94)	1129 (0.74)	
**Average exercise**				
**None**	185,261 (54.78)	87,543 (47.19)	97,718 (63.99)	<0.0001
**Seldom**	84,103 (24.87)	56,850 (30.65)	27,253 (17.85)	
**Sometimes**	35,211 (10.41)	22,357 (12.05)	12,854 (8.42)	
**Often**	9314 (2.75)	5562 (3.00)	3752 (2.46)	
**Always**	24,327 (7.19)	13,198 (7.11)	11,129 (7.29)	

^a^ From t-test for continuous variables or chi square test for categorical variables. ALT: alanine transaminase; GGT: gamma-glutamyl transpeptidase. Values expressed as mean ± SD or number (%).

**Table 2 jcm-08-01445-t002:** Hazard ratios for liver outcomes by low (30 U/L) and conventional (40 U/L) alanine aminotransferase cut-off values in men and women, estimated using Cox proportional hazard regression analyses and adjusted for age, body mass index, smoking, exercise, alcohol consumption, fasting blood glucose, and cholesterol.

Events	Men				Women			
	Low ALT (<30) ^a^		Conventional ALT (>40) ^b^		Low ALT (<19) ^c^		Conventional ALT (>40) ^d^	
	HR	95% CI	HR	95% CI	HR	95% CI	HR	95% CI
**Liver-related mortality**	5.18	(4.04, 6.63)	6.96	(5.48, 8.86)	4.78	(2.76, 8.27)	12.06	(7.43, 19.58)
**Hepatocellular carcinoma**	1.45	(1.40, 1.49)	1.67	(1.60, 1.74)	1.12	(1.08, 1.17)	1.70	(1.57, 1.83)
**Decompensated liver events**	3.42	(2.95, 3.97)	4.43	(3.80, 5.17)	1.85	(1.45, 2.35)	4.29	(3.17, 5.78)

^a^ Criteria to distinguish between normal and abnormal. Above 30 is abnormal and below 30 is normal. ^b^ Cut-off values of ALT values that discriminate between normal and abnormal in men. Above 40 is abnormal, and below 40 is normal. ^c^ Cut-off values of ALT values that discriminate between normal and abnormal in men. Above 19 is abnormal and below 19 is normal. ^d^ Cut-off values of ALT values that discriminate between normal and abnormal in men. Above 40 is abnormal and below 40 is normal. ALT: alanine transaminase, HR: Hazard ratios, CI: confidence interval.

**Table 3 jcm-08-01445-t003:** Concordance statistic (C-index) value of each cutoff value in men and women.

Criteria	Men (*n* = 185,510)				Women (*n* = 152,706)			
	Low ALT (<30 U/L) ^a^ (*n* = 54,488)		Conventional ALT (> 40 U/L) ^b^ (*n* = 24,881)		Low ALT (<19 U/L) ^c^ (*n* = 62,658)		Conventional ALT (>40 U/L) ^d^ (*n* = 6942)	
	C-Index ^c^	95% C.I.	C-Index ^c^	95% C.I.	C-Index ^c^	95% C.I.	C-Index ^c^	95% C.I.
**Liver-related mortality**	0.81	(0.77, 0.84)	0.81	(0.78, 0.84)	0.86	(0.79, 0.92)	0.88	(0.81, 0.94)
**Hepatocellular carcinoma**	0.74	(0.74, 0.75)	0.74	(0.74, 0.75)	0.73	(0.73, 0.74)	0.73	(0.73, 0.74)
**Decompensated liver events**	0.73	(0.71, 0.75)	0.73	(0.71, 0.75)	0.73	(0.69, 0.76)	0.74	(0.71, 0.77)

^a,b^ Cutoff values of ALT values that discriminate between normal and abnormal in men and women. ^c^ Estimated using Cox proportional hazard regression analysis. Adjusted for age, BMI, smoking, exercise, alcohol consumption, fasting blood glucose, and cholesterol. ALT: alanine transaminase, C-index: Concordance statistic.

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
