# Peer review of "Low Alanine Aminotransferase Cut-Off for Predicting Liver Outcomes; A Nationwide Population-Based Longitudinal Cohort Study"

_jcm, 2019, doi:10.3390/jcm8091445_

Round 1
Reviewer 1 Report
Line 77 to 80: This section is without punctuation mark. I suggest you edit this section to make it shorter so its easier to understand. Also what is liver related unfavorable outcomes? Perhaps a few examples to make it clear on what you are aiming to investigate.
Line 85: In South Korea,
Line 85 to 93: Is this necessary when you are only using 1 set of database?
Line 94 to 96: lacking punctuation marks.
Line 96: typo.'who had"
Line 98, punctuation mistake
Line 127: typo, IN
I find figure 3 confusing, it does not look like a Kaplan-Meier cureves, perhaphs not the most appropriate with dash line. possible to have different colour? Also having this results in a table form is probably more meaningful. At present I do not think a non stats expert would be able to draw conclusion from the figure.
Discussion
Line 17: typo.
Line 35,39: referencing error
Reviewer 2 Report
One of the biggest concern I have with this paper is lack of reference. It is very difficult to cross check their claims. Because there is no reference.
In introduction, if there is any data that should be supported by reference. Good amount of revision should be there in introduction.
Author Response
One of the biggest concern I have with this paper is lack of reference. It is very difficult to cross check their claims. Because there is no references.
In introduction, if there is any data that should be supported by reference. Good amount of revision should be there in introduction.
☞ Thank you for your comments. The reference part seems to be missing when updating the data. We are sorry to have confused you because of our mistake. As pointed out, we have added references and data to support the content of the introduction and discussion.
Round 2
Reviewer 2 Report
NA